# The Roles of Sex Hormones in the Course of Atopic Dermatitis

**DOI:** 10.3390/ijms20194660

**Published:** 2019-09-20

**Authors:** Naoko Kanda, Toshihiko Hoashi, Hidehisa Saeki

**Affiliations:** 1Department of Dermatology, Nippon Medical School, Chiba Hokusoh Hospital, Inzai, Chiba 270-1694, Japan; 2Department of Dermatology, Nippon Medical School, Bunkyo-Ku, Tokyo 113-8602, Japan; t-hoashi@nms.ac.jp (T.H.); h-saeki@nms.ac.jp (H.S.)

**Keywords:** atopic dermatitis, estrogen, progesterone, androgen, dehydroepiandrosterone, T helper 2 cell, skin barrier

## Abstract

Atopic dermatitis (AD) is a chronic inflammatory skin disease characterized by T helper 2 cell (Th2)-shifted abnormal immunity, skin barrier impairment, and pruritus. The prevalence of AD in childhood is slightly higher in boys than in girls; after puberty, the sexual difference is reversed. The female preponderance in all generations exists in intrinsic AD with enhanced Th1 activity and nickel allergy, lacking increased serum IgE or filaggrin mutation. AD is often deteriorated before menstruation. We review the effects of sex hormones on immune responses and skin permeability barrier and propose possible hypotheses for the above phenomena. After puberty, the immune responses of patients are remarkably influenced by sex hormones. Estrogen and progesterone enhance the activities of Th2/regulatory T cell (Treg) but suppress Th1/Th17. Androgens suppress Th1/Th2/Th17 and induce Treg. The skin permeability barrier is fortified by estrogen but is impaired by progesterone and androgens. Dehydroepiandrosterone suppresses Th2 but enhances Th1. The amount of steroid sulfatase converting dehydroepiandrosterone sulfate to dehydroepiandrosterone is higher in women than in men, and thus, women might be more susceptible to the influence of dehydroepiandrosterone. The balance of modulatory effects of sex hormones on immune responses and skin barrier might regulate the course of AD.

## 1. Introduction

Atopic dermatitis (AD) is a chronic inflammatory skin disease characterized by T helper 2 cell (Th2)-shifted abnormal immunity, skin barrier impairment, and pruritus (Figure 1) [1,2,3]. These three elements are mutually related and organize the clinicopathological features of AD. AD patients mostly show reduction of filaggrin expression partly due to mutation of this gene [1,2,3] and reveal decreases in water content and of ceramide synthesis in the stratum corneum (SC) [1,2,3]. Moreover, tight junctions (TJs) are dysfunctional in AD: the levels of zonula occludens 1 were decreased in the non-lesional sites of AD, and the levels of zonula occludens 1 and claudin-1 were decreased in the lesional sites relative to the levels in skin from healthy subjects [4]. Such impaired SC and TJ barriers allow the penetration of allergens like house dust mite, food, or pathogens, inducing sensitization to these allergens [1,2,3]. The AD lesional skin is infiltrated mainly by Th2 cells producing interleukin-4 (IL-4), IL-13, or IL-31 and by T22 cells producing IL-22, while chronic lesion is associated with the infiltration of Th1 cells producing interferon-γ (IFN-γ). Recently, the infiltration of Th17 cells is also noted in AD lesions [5]. Most AD patients show increased serum IgE levels and specific IgE antibodies against a variety of environmental allergens [1,2,3]. AD patients suffer from severe pruritus due to a variety of pruritogens like histamine, cytokines like thymic stromal lymphopoietin (TSLP), IL-31, IL-4, IL-13, or neuropeptides and abnormal extension of sensory nerves into the epidermis due to the increased expression of nerve growth factor or artemin or to the decreased expression of semaphorin 3A [2].

The prevalence of AD in childhood is slightly higher in boys than in girls: 8.7% and 5.6% for boys and girls, respectively, at <4 years old in the Netherlands [6]. After puberty, there is a slightly higher prevalence of AD in females: 5.7% and 8.1% for men and women, respectively, in Japan [7] or 6.04% and 8.01% in Europe and the USA [8]. This tendency is more remarkable in asthma, another Th2-shifted allergic disease [9]; the female:male percentage of patients was 35:65 at ages of 2 to 13 years, was inversed with 65:35 at ages of 23 to 64 years, are similar between those at ages of 14 to 22 years [10]. Adult female patients with asthma show more severe symptoms than adult male patients; the percentage of hospitalization for symptoms is 68% versus 32% in females versus males [11].

After puberty, the secretion of sex hormones from the ovary, testis, or adrenal gland is enormously increased. The immune responses or skin barrier in adolescents and adults might thus be more susceptible to influence by sex hormones compared to those in childhood. The effects of sex hormones might be related to the generation-dependent sexual difference in the prevalence of allergic diseases [12]. Interestingly, female preponderance of AD-like dermatitis possibly after puberty is detected in KFRS4/Kyo rats [13]. Dermatitis with severe pruritus initially appeared around 4 months of age, rapidly worsened from 6 to 8 months of age, and predominantly occurred in females: 100% of female versus 50% of male KFRS4/Kyo rats of 8 months old that were examined. The skin lesions were infiltrated with eosinophils, mast cells, and lymphocytes and were associated with increased plasma IgE levels and increased Th2 and Th17 cytokine mRNA levels in the skin-draining lymph nodes. Rats become sexually mature at about the sixth week but attain social maturity 5–6 months later [14], corresponding to the age with worsening dermatitis in KFRS4/Kyo rats. It is thus hypothesized that female sex hormones like estrogen or progesterone may contribute to the higher incidence of dermatitis in female KFRS4/Kyo rats.

On the other hand, female preponderance in all generations exists in intrinsic AD with enhanced Th1 activity and high incidence of nickel (Ni) allergy and without increased serum IgE values or filaggrin mutation [15]. Moreover, in female patients with AD, the disease is often deteriorated before menstruation, i.e., in the luteal phase when both estrogen and progesterone are secreted (Figure 2) [16]. These phenomena indicate that sex hormones might modulate the course of AD in the context of immune responses, skin barrier, or pruritus.

In this article, we firstly review the previous studies regarding the regulatory effects of sex hormones on the immune responses and skin barrier. We next propose hypotheses on possible hormonal regulation in the generation-dependent sexual difference in the prevalence of extrinsic AD, female preponderance of intrinsic AD, and premenstrual deterioration of AD.

## 2. The Effects of Sex Hormones on Immune Responses

### 2.1. General Tendency (Table 1)

Female hormones estrogen and progesterone mostly enhance the activities of Th2 cells and regulatory T cells (Tregs) but suppress Th1 and Th17 activities, which is favorable for the acceptance of allogeneic fetus during pregnancy [17]. Androgens like testosterone or dihydrotestosterone (DHT) are mostly immunosuppressive and suppress Th1, Th2, and Th17 activities but induce Treg activity. The magnitude of stimulation or suppression by female hormones is mostly higher than that by male hormones [18]. Generally in adolescents and adults, Th1 activities are higher in men than in women while Th2 activities are much higher in women than in men [18]. The sexual differences in Th17 or Treg activities are ambiguous. Dehydroepiandrosterone (DHEA) produced in the adrenal cortex enhances Th1 responses and shifts the balance of Th1/Th2 toward Th1-dominant immunity [19]. Females have higher amounts of steroid sulfatase converting dehydroepiandrosterone sulfate (DHEAS) to active DHEA and, thus, might be more susceptible to influence by DHEA compared to males [20]. To date, it has not been precisely examined how sex hormones regulate the activity of Th22 cells producing IL-22 alone without IL-17A.

### 2.2. Female Hormones

#### 2.2.1. Estrogens (Table 2)

Estrogens are estrone, estradiol (E2), and estriol. E2 is produced by ovarian granulosa cells and placenta. The immunomodulatory effects of E2 are mediated mainly by intracellular estrogen receptor α (ERα) and ERβ or structurally unrelated membrane G-protein-coupled estrogen receptor 1. E2 promotes Th2 activity [26,32]: E2 at pregnancy levels of concentration enhanced the expression of GATA binding protein 3 (GATA3) and IL-4 in ovariectomized experimental autoimmune encephalomyelitis (EAE) model mice [23]. E2 also enhanced IgE production in mouse splenocytes [27].

The effects of E2 on Th1 cells are complex and depend on the concentration, tissue, or disease context: E2 at estrous levels of concentration in vivo enhanced the expression of T-box-containing protein expressed in T cell (T-bet) and IFN-γ production in female ovariectomized autoimmune thyroiditis model mice [21]. In contrast, E2 at pregnancy levels of concentration reduced the production of IL-12 and IFN-γ in phytohemagglutinin plus lipopolysaccharide (LPS)-stimulated human whole blood cells [34]. The high-dose estrogen treatment reduced the expression of IFN-γ and T-bet in EAE model mice [23]. It appears that pregnancy levels of E2 shift the Th1/Th2 balance towards Th2 profiles, inhibiting Th1 development [22,35].

Though E2 mostly suppresses Th17 activity [22,24], several studies reported the stimulatory effects of E2 on Th17 cells: E2 at estrous levels of concentration enhanced the expression of IL-21 and retinoic acid receptor-related orphan nuclear receptor γt (RORγt) in female ovariectomized autoimmune thyroiditis model mice [21]. Diarylpropionitrile, a specific agonist of ERβ and not of ERα, in vivo enhanced IL-17A, IL-21, and RORγt mRNA levels in splenocytes of experimental autoimmune thyroiditis model mice through binding of the agonist-activated ERβ to *IL-17A* and *IL-21* gene promoters [29]. In contrast, E2 at estrous levels of concentration suppressed RORc expression and IL-17A and IL-22 production in response to sperm or *Candida albicans* in female ovariectomized mice [25]. The treatment with estrogen at pregnancy levels of concentration suppressed RORγt expression and IL-17A and IL-6 production in EAE model mice [23]. E2 upregulated the expression of repressor of estrogen receptor activity (REA) and recruited REA to the estrogen response elements (EREs) on the *RORγt* promoter region, thus inhibiting RORγt expression [36].

E2 enhances the activity and/or proliferation of Tregs [28]: E2 enhanced the expression of forkhead box P3 (Foxp3) by inducing binding of the E2/ERα complex to EREs on a human *Foxp3* promoter [30]. Polanczyk et al. reported that E2 in vivo and in vitro increased Foxp3 expression and Treg number in mice [31]. Tai et al. reported that E2, at physiological doses, in vitro stimulated the conversion of CD4+CD25-T cells into CD4+CD25+T cells, which exhibited the enhanced expression of Foxp3 and IL-10 in mice [12].

Estrogen also acts on mast cells and induces IgE-mediated degranulation [33,37], indicating the stimulatory effects of E2 on allergic diseases. In Th2 hapten toluene diisocyanate (TDI)-sensitized allergic airway inflammation model mice, ERα and ERβ agonists induced IL-33 production of airway epithelial cells and eosinophil infiltration into the lung [38].

#### 2.2.2. Progesterone (Table 3)

Progesterone is secreted by the ovarian corpus luteum and placenta and plays a major role in the establishment and maintenance of pregnancy. The effects of progesterone are mainly mediated by intracellular progesterone receptors while some rapid non-transcriptional actions are mediated by structurally unrelated membrane progesterone receptors [17].

Progesterone promotes Th2 activity [45]: progesterone acts on T cells and induces the secretion of progesterone-inducible blocking factor (PIBF) which binds IL-4 receptor α(IL-4Rα)/PIBFR on the cell surface and induces the Janus kinase 1 (Jak1)/signal transducer and activator of transcription 6 (STAT6) pathway to increase the production of Th2 cytokines like IL-4 or IL-10 [40]. Progesterone increased TSLP expression in vaginal epithelium and GATA-3 expression and IL-4 production in CD4+T cells in *Neisseria gonorhoeae*-infected vagina of mice [41]. Progesterone treatment increased IL-4 production in peripheral blood mononuclear cells (PBMCs) from pregnant cows [39]. The pro-Th2 effect of progesterone is consistent with the higher IL-4 and IL-10 production in PBMCs from pregnant cows with high progesterone levels than those from nonpregnant cows [39]. Progesterone treatment on ovariectomized asthma model mice increased serum IgE levels and IL-4 production in bronchoalveolar lavage cells [42], indicating the contribution of progesterone to the elicitation of allergic diseases.

In contrast, progesterone directly suppresses Th1 development in mice [46]. Progesterone treatment suppressed T-bet expression and IFN-γ production in PBMCs from pregnant cows [39]. The Th1-suppressive effect of progesterone is also consistent with the decreased IFN-γ production in PBMCs from pregnant cows compared to those from nonpregnant cows [39].

Progesterone suppresses the differentiation of Th17 cells: progesterone in vitro suppressed IL-17A, IL-17F, and IL-21 production and RORc expression in human cord blood cells under the Th17-differentiation conditions and also suppressed their STAT3 phosphorylation in response to IL-6 [43]. Progesterone suppressed RORγt expression and decreased IL-17A-producing CD4+T cell numbers in *Neisseria gonorhoeae*-infected vagina of mice [41].

Progesterone induces the differentiation of Tregs [44]: progesterone in vitro drove the allogeneic activation-induced differentiation of human cord blood naive T cells into immunosuppressive Tregs, which highly expressed FoxP3 and memory T cell marker CD45RO [43]. Progesterone increased the percentage of CD4+CD25+Foxp3+ Tregs in *Neisseria gonorhoeae*-infected vagina of mice [41]. These reports totally suggest that progesterone favors Th2/Treg activities but suppresses Th1/Th17 activities, which might be favorable for tolerance to allogeneic fetus and for maintenance of pregnancy [17].

### 2.3. Androgens (Table 4)

Androgens, such as dihydrotestosterone (DHT) or testosterone, are synthesized in the gonads and adrenal glands. Testosterone is the most concentrated androgen in adult male serum. DHT is present at one-tenth the concentration of testosterone though DHT is more potent than testosterone. Testosterone can be aromatized to E2 by aromatase. Androgens mainly bind intracellular androgen receptors (ARs) but also bind plasma membrane G-protein-coupled receptors [56].

Androgens are mostly immunosuppressive [51,57]. Androgens inhibit Th1 differentiation: testosterone inhibited IL-12-induced phosphorylation of STAT4 in murine CD4+T cells by inducing the expression of protein tyrosine phosphatase nonreceptor 1, which inactivates Jak2 and Tyk2 kinases [47]. DHT inhibited IFN-γ production in murine CD4+T cells by enhancing the expression of peroxisome proliferator-activated receptor α (PPARα), which represses *IFNγ* transcription [48].

Androgens inhibit Th2 differentiation: DHT treatment of bone-marrow-derived dendritic cells pulsed with *Trichuris muris* antigen reduced their Th2-priming activity with a complete ablation of IL-4, IL-10, and IL-13 production by co-cultured T cells [54]. DHT treatment of prostate stromal cells suppressed the production of IL-4, IL-5, and IL-13 production by co-cultured CD4+ T cells [50]. Gonadectomized male mice showed increased IL-13-producing innate lymphoid cell 2 (ILC2) and Th2 cells and increased serum IgE levels compared to sham-operated mice in response to house dust mite antigens [53], indicating the suppressive effects of androgens on type 2 immune responses. In male castrated phospholipase A-sensitized allergic rhinitis model mice, testosterone administration decreased the production of phospholipase A-specific IgE [52], indicating the suppressive effects of androgens on allergic rhinitis. Androgens also suppress B cell lymphopoiesis [51]: testosterone acted on bone marrow stromal cells to induce the production of transforming growth factor-β, which reduced the production of IL-7 required for B cell proliferation and differentiation [56].

Androgens suppress Th17 differentiation: male gonadectomized mice showed increased IL-17A-producing Th17 and γδT cells compared to sham-operated mice in response to house dust mite antigens [53]. The deficiency of AR signaling enhanced IL-17A production and IL-23R expression in T cells under Th17-differentiation conditions [53]. These results indicate the inhibitory effects of androgens on type 17 immune responses. Testosterone treatment of murine T cells in vitro decreased IFN-γ or IL-17 production under the Th1- or Th17-differentiation conditions, respectively [49].

On the other hand, androgens induce Tregs: androgen-activated AR bound androgen response elements on the *Foxp3* promoter and enhanced the acetylation of histone H4 on the promoter, allowing the binding of additional transcription factors, and thus enhanced the expression of Foxp3 in human T cells [55]. Danazol, an attenuated androgen, increased CD4+CD25highCD127lowFoxp3+ Tregs in patients with aplastic anemia [58].

### 2.4. DHEA (Table 5)

DHEAS is the most abundant steroid in human blood serum and is secreted by the adrenal cortex [69]. DHEAS is converted by steroid sulfatase to an active form, DHEA [69]. Steroid sulfatase is controlled by an X-linked gene that escapes the Lyon effect of X-inactivation; as a result, women have twice the amount of steroid sulfatase than men [70], especially in peripheral lymphoid organs [20]. The DHEA/DHEAS ratio in circulation is usually higher in women than in men [71], and under 50 years old, the plasma DHEA concentration of women is higher than that of men [72]. It is thus hypothesized that women might be more susceptible to the effects of DHEA than men [20] though the effects might be influenced by other factors such as receptor and downstream pathways. DHEA itself binds nuclear steroid hormone receptors like AR, ERα, or ERβ with lower affinities than cognate ligands. DHEA is metabolized to other steroid hormones, testosterone, DHT, or E2 (Figure 3), and these metabolites bind the corresponding steroid receptors: DHEA metabolite 5-androsten-3β,17β-diol can bind ERβ [61]. Moreover, recent studies revealed that DHEA and DHEAS act as ligands of many nuclear receptors like PPARα, pregnane X receptor, and constitutive androstanol receptor and membrane receptors like TrkA, N-methyl-D-aspartic acid receptors, or γ-amino butyric acid receptors [73]. Thus, the biological effects of DHEA depend on the levels of metabolizing enzymes and individual receptors and, thus, vary with species, tissues, or cell types.

DHEA mostly enhances Th1 differentiation: DHEA treatment enhanced IL-12 and IFN-γ production in female murine peritoneal cells in vivo and enhanced the expression of very late antigen-4 and leukocyte function-associated antigen-1 in CD4+ T cells in vitro [19]. DHEA in vivo increased the expression of vascular cell adhesion molecule-1 and intercellular adhesion molecule-1 in ovary granulosa cells of mice [19,74]. DHEA suppresses type 2 immune responses [65]: in female ovalbumin-sensitized asthma model mice, DHEA administration reduced eosinophil infiltration into the lung; serum ovalbumin-specific IgE levels; and the expression of IL-4, IL-5, and IL-13 and type 2 chemokines CC-chemokine ligand 1 (CCL1) and CCL24 in bronchoalveolar lavage fluid but increased IFN-γ production in ovalbumin-activated splenocytes [59]. DHEA reduced IL-4 production but increased IL-2 production in concanavalin A-stimulated human PBMCs [64]. In LPS-induced experimental inflammation model mice, DHEA increased the Th1/Th2 ratio in spleen T cells [63]. DHEA significantly increased Th1 cytokine levels (IL-2 and IFN-α) and decreased Th2 cytokine levels (IL-4 and IL-10) in primary cultured spleen T cells [63]. DHEA decreased Na^+^K^+^-ATPase activity and increased Ca^2+^ ATPase activity in T cells, which might regulate the balance of cytokine secretion [63]. Thus, DHEA enhances Th1 immune response and regulates the balance of Th1/Th2 toward Th1-dominant immunity. In AD model mice induced by topical 2,4-dinitrochlorobenzene (DNCB), oral or topical DHEA attenuated the infiltration of eosinophils and mast cells into DNCB-challenged ear skin. In those mice, oral or topical DHEA also reduced serum IL-4 and IgE levels and reduced IL-4 and IL-5 production but increased IFN-γ production in splenocytes [60]. DHEA in vitro suppressed the production of type 2 chemokines, CCL17 and CCL22, in tumor necrosis factor-α-stimulated HaCat cells [60].

The effects of DHEA on Th17 and Treg are ambiguous [62] and dependent on tissue or disease context [66]: DHEA treatment increased the number of Foxp3+ Tregs in splenocytes of female ovariectomized mice [75]. In EAE model mice and human multiple sclerosis patients, DHEA directly inhibited the activity of Th17 cells, inducing IL-10-producing Tregs via ERβ activation [61]. This effect may be mediated by 5-androsten-3β,17β-diol, converted from DHEA by 17β-hydroxysteroid dehydrogenase (Figure 3). Synthetic DHEA analog HE3286 increased the frequency of CD4+CD25+Foxp3+ Tregs in spleen in collagen-induced arthritis model mice [68] though the receptors for this analog are unknown and not ERα, ERβ, or AR. In contrast, DHEA reduced the expression of Foxp3 without altering Treg frequency in PBMCs from patients co-infected with HIV and tuberculosis [67].

## 3. The Effects of Sex Hormones on the Skin Barrier (Table 1)

Overall, in adults, skin hydration is slightly higher in women than in men and basal transepidermal water loss (TEWL) is significantly higher in men than in women [35,76]. The epidermal permeability barrier is impaired by androgens and progesterone but is restored by E2 [77]. E2 paradoxically worsens the progesterone-induced barrier impairment [78]. The skin barrier is impaired in the luteal phase when both progesterone and estrogen are secreted [79].

The ovariectomy of female C57/BL6 mice reduced skin hydration, delayed skin permeability recovery after tape stripping, and reduced epidermal thickness; these were restored by the administration of E2 [80]. Ovariectomy also reduced the expression of desmoglein-1, involucrin, and loricrin, key constituents of corneodesmosome and cornified envelope (CE) in SC; these were restored by E2 treatment [80]. Ovariectomy of female nude mice (ICR-Foxn1nu) also reduced the expression of filaggrin and integrin β in the skin [81]. These results support that E2 strengthens the skin permeability barrier and integrity.

The castration of male mice (Sk:h1) or treatment with AR antagonist flutamide accelerated skin permeability recovery after tape stripping, and testosterone administration delayed recovery in the castrated mice [82]. Testosterone treatment of castrated mice reduced the number of lamellar bodies in the cytosol of stratum granulosum (SG) cells and decreased secreted lamellar contents at the interface of SC and SG [82]. These results support that testosterone perturbs the skin permeability barrier homeostasis. The administration of E2 in pregnant rats reduced TEWL of day-20 fetus while that was increased by DHT [77]. The administration of E2 accelerated lipid deposition and formation of lamellar unit structures in the SC of day-20 fetal epidermis in utero while these were delayed by DHT [77]. In fetal skin explants from day-17 rats, estrogen increased while DHT reduced the activity of β-glucocerebrosidase, which converts glucosylceramides to ceramides and is essential for the formation of lamellar unit structures throughout SC interstices [83]. In the same explants, estrogen accelerated while DHT delayed the expression of filaggrin and loricrin, key constituents of keratohyalin granules and CE, respectively [84]. These results indicate that E2 accelerates while androgen delays SC barrier formation.

Topical testosterone delayed skin permeability barrier recovery in male hairless mice (HR-1), and the delay was overcome by co-application of E2 [78]. Progesterone also delayed the recovery; however, the delay was paradoxically enhanced by E2 [78]. To date, the precise mechanism of how progesterone delays skin permeability barrier recovery is not elucidated; however, it is indicated that progesterone opposes the protective effect of E2 on skin permeability barrier and that the skin permeability barrier may be impaired in females in the luteal phase when both progesterone and estrogen are secreted. Muizzuddin et al. showed that the skin permeability barrier was the weakest between days 22 and 26 of the menstrual cycle, the mid-luteal phase [85]. TEWL is higher at the day of minimal estrogen/progesterone ratio (around day 26) than that at the day of maximal estrogen secretion (around day 13) [79]. It is thus hypothesized that the skin barrier might be protected by E2, which might be opposed by progesterone. The skin patch test reaction to Ni is higher at the luteal phase than that at the ovulatory phase or at the follicular phase [86], which may be due to the skin permeability barrier impairment at the luteal phase when progesterone is secreted. In the uterus of ovariectomized mice, E2 treatment increased mRNA levels of small proline-rich protein 2 (SPRR2), which was dampened by progesterone [87]. Since SPRRs are key constituents of CE in SC of the skin, further studies should elucidate if such suppressive effects of progesterone might be reproduced in the skin. In contrast, progesterone upregulated the expression of TJ proteins occludin and zona occludens 2 in the epidermis, and this effect was canceled by E2 in ovariectomized FvB mice [88]. Progesterone upregulated the expression of occludin in human gut tissues and Caco-2 cells [89]. It is thus hypothesized that the TJ barrier might be restored by progesterone but impaired by E2, opposite to the SC barrier, though confirmative studies are required for its verification.

It is reported that oral DHEA in humans at > 60 years old improved skin hydration, epidermal thickness, sebum production, and pigmentation [90]. Such antiaging effects of DHEA are mainly generated by the enhanced collagen biosynthesis and deposition: topical DHEA increased the mRNA levels of procollagen 1/3 and heat shock protein 47, a type 1 collagen chaperone protein in human dermal fibroblasts [91]. In contrast, topical DHEA treatment reduced the expression of genes associated with the terminal differentiation and cornification of keratinocytes, corneodesmosin, claudin 8, SPRR2G, late envelope protein 7, and Jagged 1, indicating the suppression of skin barrier properties [92]. To date, the direct effects of DHEA on skin permeability barrier have not been reported and should further be examined. In the testis, DHEAS but not DHEA augmented the TJ connections between Sertoli cells by promoting the expression of claudin-3 and -5 via membrane Gn_α11_-coupled receptors independent of AR [93].

## 4. The Effects of Sex Hormones on the Pruritus

There have been no reports that sex hormones act as pruritogens. However, E2 and/or progesterone directly or indirectly induce the secretion of Th2-related cytokines, IL-4, IL-13, IL-31, TSLP, or IL-33; these cytokines bind the corresponding receptors on C-type sensory neurons and generate an itch sensation [94]. In Th2 hapten TDI-sentitized allergic dermatitis model mice, oral or topical administration of ERα agonist propylpyrazoletriol induced TSLP and IL-33 expression in keratinocytes and promoted scratching behavior [95]; the pruritus might be caused by TSLP and IL-33. Estrogen also acted on mast cells and promoted the release of histamines [33,37], which also act as pruritogens.

In sensory neuron-keratinocyte coculture model, DHEA produced by keratinocytes promoted neurite growth possibly through the activation of TrkA [96,97], indicating the relations to abnormal neurite outgrowth into the epidermis of AD lesions. The effects of sex hormones on neurite growth should further be examined precisely.

## 5. Intrinsic AD

Intrinsic AD patients occupy around 10–20% of whole AD, show normal IgE values, lack IgE antibodies against environmental or food allergens, and lack barrier disruption and filaggrin gene mutation [15]. Intrinsic AD patients manifest Dennie–Morgan fold but without icthyosis vulgaris and palmar hyperlinearity and with milder severity of AD. Intrinsic AD patients show positive on patch tests to metals, especially to Ni, at a higher percentage than extrinsic AD patients, indicating the higher incidence of metal allergy [98]. Intrinsic AD patients show high Ni concentrations in serum and sweat [98,99], indicating the enhanced absorption and/or transport of Ni derived from food and sensitization with Ni in the circulation and skin. Immunologically, intrinsic AD patients show higher Th1 activity than that of extrinsic AD patients and show Th2 activity comparable to that of extrinsic AD patients [100]. Since Ni interacts with toll-like receptor 4 in addition to major histocompatibility/self-peptide complex and induces a mixed Th1 and Th2 cytokine responses [98,101,102,103], a considerable number of intrinsic AD patients, though not all, might show both Th1 and Th2 responses to Ni through its presentation by antigen presenting cells in the skin patch test. Thus, metals might act as the main allergens in intrinsic AD though other agents might also work in its pathogenesis. In both children or adults, the prevalence of intrinsic AD is higher in females than in males [104,105]; moreover, the prevalence of Ni allergy is higher in females than in males [106,107].

## 6. Possible Hypotheses on the Generation-Dependent Sexual Difference in the Prevalence of Extrinsic AD (Table 6)

It is hypothesized that prepubertal children might be mostly devoid of the influence of sex hormones considering their very low concentrations. The effects of DHEA, shifting the Th1/Th2 balance to Th1, might be more remarkably revealed in girls than in boys due to the higher levels of steroid sulfatase converting DHEAS to DHEA. Thus, both atopic asthma and extrinsic AD, Th2-shifted allergic diseases with atopic diathesis, might be more prevalent in boys than in girls. The male preponderance in childhood asthma might also involve gender-specific factors other than hormonal regulation [108]: boys have smaller airway diameters relative to lung volume (dysanapsis; [109]) and show a higher percentage of positive skin prick tests or IgE antibodies against aeroallergens than girls, indicating higher atopic diathesis in boys than in girls [110]. The higher aeroallergen sensitivities in boys may also be related to the male predominance in childhood AD [111]. Alternatively, the higher asthma incidence in boys may be because underdiagnosed and undertreated asthma patients exist among girls [112].

After puberty, the levels of sex hormones, estrogen, progesterone, testosterone, or DHT are greatly increased and individuals might thus be more greatly influenced by the immunological effects of those sex hormones than those of DHEA [18,113,114]. Women are exposed to higher levels of estrogen and progesterone promoting Th2 activity. The prevalence and severity of atopic asthma might thus become much higher in women than in men exposed to higher levels of androgens suppressing Th2 activity. The female preponderance in adult asthma may also involve other gender-specific factors [108]: the bronchial hyperresponsiveness is more frequent in women than in men; women are more likely to be exposed to indoor aeroallergens than men.

In the case of adult extrinsic AD, patients are influenced by the effects of sex hormones both on immune responses and skin barrier impairment. The Th2 shift is more prevalent in women than in men; however, skin barrier impairment is more likely to occur in men exposed to high levels of androgens perturbing skin permeability barrier. In women, skin barrier is restored by estrogen at the follicular and ovulatory phases but are disturbed by progesterone at the luteal phase. Totally, female adults might be slightly more protected from skin barrier impairment compared to male adults, considering the postmenopausal skin barrier disruption in females and its restoration by hormonal agents, including both estrogen and progesterone [115]. Thus, female adults with higher Th2 activity but slightly more protection from skin barrier impairment may show a slightly higher prevalence of extrinsic AD compared to male adults, though the sexual difference is moderate compared to strict female preponderance of adult atopic asthma. It is known that Th1 activity is enhanced in the chronic phase of AD and that Th17 cells may also be involved in the pathogenesis of AD [5], which might be more applicable to women than men since estrogen might promote Th1 and Th17 activities at estrous level of concentrations [21]. Other gender-specific factors might also be related to the moderate female preponderance of adult AD: women are more responsive to pruritogens than men, and female mice showed higher scratching counts than male mice under the stimulus of proteinase-activated receptor-2-activating peptide [116]. The involvement of sex hormones in hyperknesis or alloknesis should further be investigated.

## 7. Possible Hypotheses on Female Preponderance of Intrinsic AD (Table 7)

We herein propose a hypothesis on female preponderance of intrinsic AD in the context of metal allergy, one possible agent of its pathogenesis. Patients with intrinsic AD are not associated with atopic diathesis, filaggrin gene mutation, or congenital skin barrier impairment and are associated with enhanced Th1 activity as well as Th2 and high incidence of Ni allergy though not in all patients. Before puberty, there is no sexual difference in the chances of Ni exposure, though in Western countries, girls are more frequently exposed to Ni due to the first pierce experiences. Th1 responses to Ni might be more remarkable in girls with higher susceptibility to DHEA promoting Th1 activity than in boys. Totally, intrinsic AD in childhood might be more prevalent in girls than in boys.

After puberty, women are more frequently exposed to Ni by wearing ornaments than men. The immune response to Ni might be gradually shifted from Th1 to Th2 since repeated elicitation with antigens alters the balance of cytokines released locally, with a shift toward Th2-dominated responses [117]. Immune responses after puberty might be more susceptible to the influence of sex hormones than of DHEA. After puberty, Th2 responses to Ni might be more remarkable in women with higher levels of estrogen and progesterone stimulating Th2 activity than men with higher levels of androgens suppressing Th2 activity. The intrinsic AD after puberty might thus become much more prevalent in women than in men. The female preponderance of intrinsic AD may also involve certain gender-dependent factors unrelated to metal allergy, which should further be identified.

## 8. Possible Hypotheses on the Premenstrual Deterioration of AD in Females

About half of female AD patients experience premenstrual deterioration of AD symptoms [118]. The deterioration might be due to the dual effects of estrogen and progesterone on Th2 activity and skin barrier. At the luteal phase, both estrogen and progesterone are secreted and, thus, Th2-skewing effects are higher than in the other phases; moreover, the skin permeability barrier is perturbed by progesterone, especially, just prior to menstruation with minimal estrogen/progesterone ratio.

The deterioration of AD during pregnancy is also reported; 52% or 61% of female AD patients who experienced pregnancy had noticed deterioration of AD during pregnancy in the UK [119] or Korea [120], respectively. The deterioration of AD during pregnancy might also reflect the effects of extremely high concentrations of E2 and progesterone on Th2 activity and skin barrier. Moreover, the prevalence of deterioration during pregnancy was higher in intrinsic AD patients (100%) compared to that in extrinsic AD patients (47.1%) in the Korean study [120]. It is thus hypothesized that intrinsic AD patients might be more susceptible to the influence of female sex hormones compared to extrinsic AD patients.

## 9. Serum Hormone Concentrations in Patients with Allergic Diseases

It is reported that serum concentrations of DHEA or testosterone are lower in male AD patients compared to the reference group [64,121]. These indicate that DHEA- or testosterone-induced suppressive effects on Th2 activity are reduced in male AD patients, which might aggravate the symptoms of AD. It is also reported that serum concentrations of DHEA or DHEAS are lower in patients with asthma or chronic spontaneous urticaria compared to the reference group, irrespective of gender [122]. It is thus speculated that the reduction of Th2-suppressive effects by DHEA might accelerate the Th2 shift in these patients though confirmatory studies are needed to verify the speculation.

It is reported that prolonged physical stress with energy and sleep deprivation reduced serum DHEA levels and increased serum DHEAS levels, indicating the decrease of steroid sulfatase activity and/or increase of sulfotransferase activity by prolonged physical stress [123]. It is thus hypothesized that the reduction of DHEA levels might be one possible cause of the stress-induced exacerbation of allergic diseases.

## 10. Conclusions

We reviewed the effects of sex hormones related to the course of AD and focused on immune responses and skin barrier. The balance of the effects of sex hormones might up- or downregulate the prevalence and course of AD. Future studies should elucidate the effects of sex hormones on Th22 activity or pruritus, using AD model mice.

## Figures and Tables

**Figure 1 ijms-20-04660-f001:**
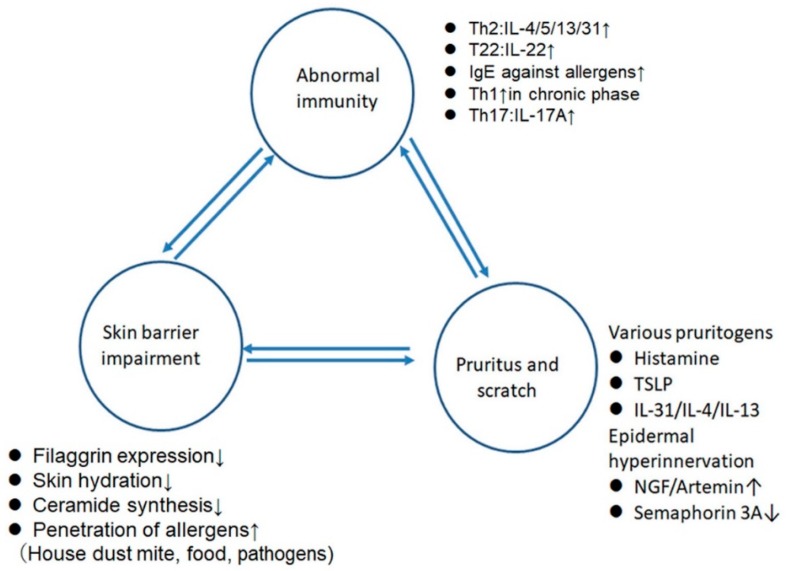
The three elements composing the pathogenesis of atopic dermatitis. Th2, T helper 2 cell; IL-4, interleukin-4; NGF, nerve growth factor; TSLP, thymic stromal lymphopoietin.

**Figure 2 ijms-20-04660-f002:**
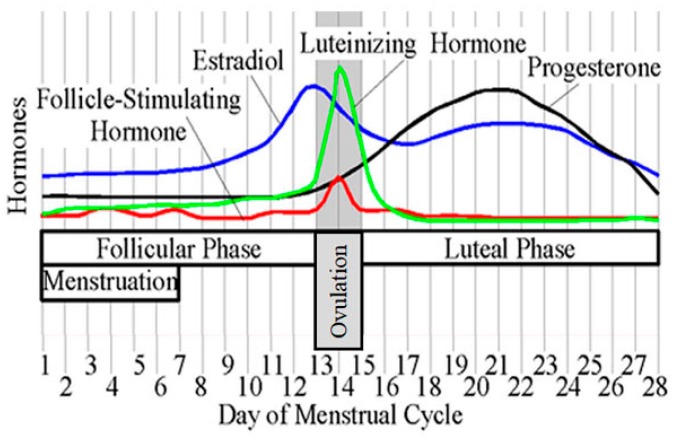
Menstrual cycle. Reprinted from https://commons.wikimedia.org/wiki/File:MenstrualCycle2.png. This file is licensed under the Creative Commons Attribution-Share Alike 3.0 Unported (https://creativecommons.org/licenses/by-sa/3.0/deed.en) license.

**Figure 3 ijms-20-04660-f003:**
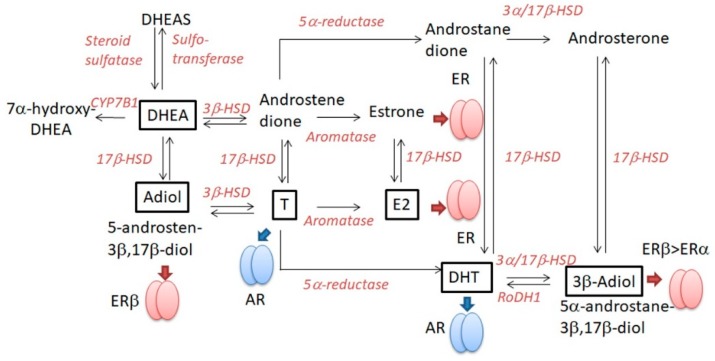
Dehydroepiandrosterone (DHEA) metabolizing pathway. DHEAS, dehydroepiandrosterone sulfate; DHT, dihydrotestosterone; T, testosterone; E2, estradiol; AR, androgen receptor; ER, estrogen receptor; 17β-HSD, 17β-hydroxysteroid dehydrogenase; RoDH1, retinol dehydrogenase type 1; ⬇, binding to the steroid receptor described.

**Table 1 ijms-20-04660-t001:** The effects of sex hormones on immune responses and skin barrier impairment.

Hormones	Th1	Th2	Th17	Treg	Skin Barrier Impairment
Androgen	↓	↓	↓	↑	↑
Estrogen	↑~⬇	⬆	↑~⬇	⬆	⬇
Progesterone	⬇	⬆	⬇	⬆	↑
DHEA	↑	↓	?	?	?
Total activity	F < M	F ≫ M	?	?	F < M

↑, Moderate stimulation; ⬆, strong stimulation; ↓, moderate suppression; ⬇, strong suppression; ?, ambiguous; ↑~⬇, Stimulatory or inhibitory effects dependent on the concentration, tissue, or disease context; Th1, T helper 1 cell; Treg, regulatory T cell; DHEA, dehydroepiandrosterone; F, female; M, male.

**Table 2 ijms-20-04660-t002:** Summary of the effects of estrogen on the activities of T helper 1 cell (Th1), Th2, Th17, and regulatory T cell (Treg).

Effects	Vivo/ Vitro	Species	Th Activities	References
Adaptive Immunity				
T-bet↑	vivo	mice	Th1↑	[21] *
IFN-γ↑	vivo	mice	Th1↑	[21] *
IL-12↑	vivo	mice	Th1↑	[21] *
T-bet↓	vitro	human	Th1↓	[22] **
T-bet↓	vivo	mice	Th1↓	[23] ^†^
IFN-γ↓	vivo	mice	Th1↓	[24] ^†^
IFN-γ↓	vitro	mice	Th1↓	[25] ^§^
IFN-γ↓	vitro	human	Th1↓	[22] **
IFN-γ↓	vivo	mice	Th1↓	[23] ^†^
GATA-3↑	vivo	mice	Th2↑	[26]
GATA-3↑	vivo	mice	Th2↑	[23] ^†^
IL-4↑	vivo	mice	Th2↑	[26]
IL-4↑	vivo	mice	Th2↑	[23] ^†^
B cells IgM, IgE↑	vitro	mice	Th2↑	[27]
RORc↓	vivo	mice	Th17↓	[24] ^†^
RORc↓	vitro	human	Th17↓	[28]
RORc↓	vitro/vivo	mice	Th17↓	[25]
RORγt↓	vitro	human	Th17↓	[22] **
RORγt↓	vivo	mice	Th17↓	[23] ^†^
IL-17A↓	vitro	mice	Th17↓	[25] ^§^
IL-17A↓	vivo	mice	Th17↓	[24] ^†^
IL-17A↓	vitro	human	Th17↓	[22] **
IL-17A↓	vivo	mice	Th17↓	[23] ^†^
IL-22↓	vitro	mice	Th17↓	[25] ^§^
IL-17A↑	vivo	mice	Th17↑	[29] ^‡^
IL-21↑	vivo	mice	Th17↑	[29] ^‡^
IL-22↑	vivo	mice	Th17↑	[21] *
RORγt↑	vivo	mice	Th17↑	[21] *
RORγt↑	vivo	mice	Th17↑	[29] ^‡^
Foxp3↑	vitro	human	Treg↑	[28]
Foxp3↑	vitro	human	Treg↑	[30]
Foxp3↑	vivo	mice	Treg↑	[23] ^†^
Foxp3↑	vivo/vitro	mice	Treg↑	[31]
IL-10↑	vivo	mice	Treg↑	[23] ^†^
IL10↑	vitro	mice	Treg↑	[12]
TGF-↑	vivo	mice	Treg↑	[23] ^†^
Foxp3↓	vitro	human	Treg↓	[22] **
IL-10↓	vitro	human	Treg↓	[22] **
Innate immunity				
Macrophage IL-10↑	vitro	human	Th2↑	[32]
Macrophage IL-1RA↑	vitro	human	Th2↑	[32]
Macrophage CD192↑	vitro	human	Th2↑	[32]
Mast cell degranulation↑	vitro	mice	Th2↑	[33]

↑, stimulation; ↓, suppression; ROR, retinoic acid receptor-related orphan nuclear receptor; Foxp3, forkhead box P3; GATA3, GATA binding protein 3; T-bet, T-box-containing protein expressed in T cells; STAT3, signal transducer and activator of transcription 3; *, Estrous level of estradiol (E2) induced the effects, pregnancy-level of E2 did not; **, Effects of polyphenolic compound delphinidin; †, Pregnancy-levels of E2; ‡, Estrogen receptor agonist diarylpropionitrile; §, 10^−10^M E2 induced the effects, but 10^−11^E2 did not; Effects of selective estrogen receptor modulator.

**Table 3 ijms-20-04660-t003:** Summary of the effects of progesterone on the activities of T helper 1 cell (Th1), Th2, Th17, and regulatory T cell (Treg).

Effects	Vivo/Vitro	Species	Th Activities	References
Adaptive Immunity				
T-bet↓	vitro	cows	Th1↓	[39]
IFN-γ↓	vitro	cows	Th1↓	[39]
PIBF- STAT6↑	vitro	human	Th2↑	[40]
GATA3↑	vivo/vitro	mice	Th2↑	[41]
IL-4↑	ex vivo	mice	Th2↑	[42]
IL-4↑	vitro	cows	Th2↑	[39]
IL-4↑	vivo	mice	Th2↑	[42]
B cell IgE↑	vivo	mice	Th2↑	[42]
Vaginal epithelial cell TSLP↑	vivo/vitro	mice	Th2↑	[41]
STAT3 RORC CCR6 IL-23R IL-6R AHR↓	vitro	human	Th17↓	[43]
RORγt↓	vivo/ vitro	mice	Th17↓	[41]
RORC↓	vitro	cows	Th17↓	[39]
IL-17A↓	vitro	human	Th17↓	[43]
IL-17A↓	vitro	cows	Th17↓	[39]
IL-17F↓	vitro	human	Th17↓	[43]
IL-21↓	vitro	human	Th17↓	[43]
CD39+ regulatory Th17↑	vivo	mice	Th17↑※	[44]
IL-17A↑	vivo	mice	Th17↑※	[44]
IL-22↑	vivo	mice	Th17↑※	[44]
IL-23↑	vivo	mice	Th17↑※	[44]
IL-6↑	vivo	mice	Th17↑※	[44]
TGF-β↑	vivo	mice	Th17↑※	[44]
Foxp3↑	vitro	mice	Treg↑	[43]
Foxp3↑	vivo/vitro	mice	Treg↑	[41]
Innate Immunity				
Airway epithelial cells Amphiregulin↑	vivo	mice		[44]

※,CD39+ regulatory Th17 cells; ↑, stimulation; ↓, suppression; ROR, retinoic acid receptor-related orphan nuclear receptor; Foxp3, forkhead box P3; GATA3, GATA binding protein 3; T-bet, T-box-containing protein expressed in T cells; STAT, signal transducer and activator of transcription; AHR, aryl hydrocarbon receptor; CCR6 CC-type chemokine receptor 6; TSLP, thymic stromal lymphopoietin; PIBF, progesterone-induced blocking factor.

**Table 4 ijms-20-04660-t004:** Summary of the effects of androgens on the activities of T helper 1 cell (Th1), Th2, Th17, and regulatory T cell (Treg).

Effects	Vivo/Vitro	Species	Th Activities	References
Adaptive Immunity				
ptpn1↑ STAT4↓	vivo/ vitro	human and mice	Th1↓	[47]
PPARα↑	vitro/vivo	mice	Th1↓	[48]
IFN-γ↓	vitro/vivo	mice	Th1↓	[48]
IFN-γ↓	vitro	mice	Th1↓	[49]
IFN-γ↓	vitro	human	Th1↓	[50]
IL-12↓	vitro	human	Th1↓	[50]
CXCL10↓	vitro	human	Th1↓	[50]
IL-13↓	vitro	human	Th2↓	[50]
IL-4↓	vitro	human	Th2↓	[50]
IL-5↓	vitro	human	Th2↓	[50]
B cell number↓	vivo	mice	Th2↓	[51]
B cell Antigen-specific IgE production ↓	vivo	mice	Th2↓	[52]
PPARγ↓	vitro/vivo	mice	Th17↑	[48]
IL-17A↑	vitro/vivo	mice	Th17↑	[48]
IL-23R↓	vivo	mice	Th17↓	[53]
IL-23R↓	vitro	Mice	Th17↓	[54]
IL-17A↓	vitro	mice	Th17↓	[49]
IL-17A↓	vivo	mice	Th17↓	[53]
IL-17A↓	vitro	human	Th17↓	[50]
ARE-Foxp3↑	vitro	human	Treg↑	[55]
IL-10↑	vitro	human	Treg↑	[50]
Innate Immunity				
Mast cell IL-4↓	vivo	mice	Th2↓	[53]
ILC2 IL-13↓	vivo	mice	Th2↓	[53]
Basophil IL-4↓	vivo	mice	Th2↓	[53]

↑, stimulation; ↓, suppression; ROR, retinoic acid receptor-related orphan nuclear receptor; Foxp3, forkhead box P3; STAT, signal transducer and activator of transcription; ptpn1, protein tyrosine phosphatase, non-receptor type 1; PPAR peroxisome proliferator-activated receptor; ARE, androgen response element; ILC, innate lymphoid cell.

**Table 5 ijms-20-04660-t005:** Summary of the effects of dehydroepiandrosterone (DHEA) on the activities of T helper 1 cell (Th1), Th2, Th17, and regulatory T cell (Treg).

Effects	Vivo/Vitro	Species	Th Activities	References
Adaptive Immunity				
IFN-γ↑	vivo	mice	Th1↑	[59]
IFN-γ↑	ex vivo	mice	Th1↑	[60]
IFN-γ↑	vitro	mice	Th1↑	[19]
IL-12↑	vitro	mice	Th1↑	[10]
IFN-γ↓	vivo	mice	Th1↓	[61] *
IFN-γ↓	vivo	mice	Th1↓	[62]
IL-4↓	vivo	mice	Th2↓	[59]
IL-4↓	vitro	mice	Th2↓	[63]
IL-4↓	vitro	human	Th2↓	[64]
IL-4↓	vivo	mice	Th2↓	[61] *
IL-4↓	vivo	mice	Th2↓	[60]
IL-4↓	ex vivo	mice	Th2↓	[60]
IL-5↓	vivo	mice	Th2↓	[59]
IL-5↓	vivo	mice	Th2↓	[62]
IL-5↓	vitro	human	Th2↓	[64]
IL-5↓	ex vivo	mice	Th2↓	[60]
IL-13↓	vivo	mice	Th2↓	[59]
CCL11↓	vivo	mice	Th2↓	[59]
CCL24↓	vivo	mice	Th2↓	[59]
B cell IgE ↓	vivo	mice	Th2↓	[65]
B cell IgE ↓	vivo	mice	Th2↓	[60]
B cell IgG1↓	vivo	mice		[60]
IL-13↑	vivo	mice	Th2↑	[62]
RORC↓	vivo	mice	Th17↓	[61] *
IL-17A↓	vivo	mice	Th17↓	[61] *
IL-17A↓	vivo	mice	Th17↓	[62]
TNF-α↓	vivo	mice	Th17↓	[60]
IL-6↓	vivo	mice	Th17↓	[62]
TGFβ↓	vivo	mice	Th17↓	[62]
TNF-α↑	vitro	human	Th17↑	[66] ^†^
IL-6↑	vitro	human	Th17↑	[66] ^†^
IL-1β↑	vitro	human	Th17↑	[66] ^†^
Foxp3↓	vitro	human	Treg↓	[67]
IL-10↓	vitro	mice	Treg↓	[62]
IL-10↓	vitro	mice	Treg↓	[63]
Foxp3↑	vivo	mice	Treg↑	[68] **
IL-10↑	vivo	mice	Treg↑	[61] *
Innate Immunity				
Mast cell infiltration↓	vivo	mice	Th2↓	[60]
Eosinophil infiltration↓	vivo	mice	Th2↓	[59]
Eosinophil infiltration↓	vivo	mice	Th2↓	[60]
HaCat cells CCL17↓	vitro	human	Th2↓	[60]
HaCat cells CCL22↓	vitro	human	Th2↓	[60]
BEAS-2B CCL11↓	vitro	human	Th2↓	[59]
BEAS-2B CCL24↓	vitro	human	Th2↓	[59]
Ovary granulosa cell ICAM1/VCAM1↑	vivo	mice		[19]

*, Possible effects of DHEA metabolite, 5-androsten-3β,17β-diol (adiol), via estrogen receptor β (ERβ); **, Synthetic DHEA analog HE3286; †, Effects via ER; ↑, stimulation; ↓, suppression; ?, ambiguous; ROR, retinoic acid receptor-related orphan nuclear receptor; Foxp3, forkhead box P3; ICAM1, intercellular adhesion molecule 1; VCAM1, vascular cell adhesion molecule 1.

**Table 6 ijms-20-04660-t006:** The generation-dependent sexual difference in the prevalence of atopic asthma and extrinsic atopic dermatitis (AD).

	**Child**			**Adolescent–Adult**		
**Atopic Asthma**	**Atopic Diathesis**		**Th2 Regulation by DHEA**	**Th2 Regulation by Sex Hormones**	**Th2 Regulation by DHEA**	
M	+~++		↓	↓↓by A	↓	
F	+		↓↓	↑↑↑↑by E, P	↓↓	
Prevalence	M > F			M ≪ F		
**Extrinsic AD**	**Atopic Diathesis**	**Filaggrin Gene Nutation**	**Th2 Regulation by DHEA**	**Th2 Regulation by Sex Hormones**	**Th2 Regulation by DHEA**	**Skin Barrier Impairment by Sex Hormones**
M	+~++	+ or −	↓	↓↓by A	↓	↑by A
F	+	+ or −	↓↓	↑↑↑↑by E, P	↓↓	↓↓by E ↑by P Totally↓
Prevalence	M > F			M < F		

↑, Stimulation; ↓, Suppression; Th2, T helper 2 cell; DHEA, dehydroepiandrosterone; F, female; M, male; A, androgen; E, estrogen; P, progesterone.

**Table 7 ijms-20-04660-t007:** Female preponderance of intrinsic atopic dermatitis (AD).

Intrinsic AD	Child				Adolescent–Adult		
	Atopic Diathesis	Filaggrin Gene Nutation	Exposure to Ni	Stimulation of Th1 Response to Ni by DHEA	Exposure to Ni	Regulation of Th2 Response to Ni by Sex Hormones	Stimulation of Th1 Response to Ni by DHEA
M	−	−	+	↑	+	↓↓by A	↑
F	−	−	+~++	↑↑	++	↑↑↑↑by E, P	↑↑
Prevalence	M < F				M ≪ F		

↑, Stimulation; ↓, Suppression; Th1, T helper 1 cell; DHEA, dehydroepiandrosterone; F, female; M, male; A, androgen; E, estrogen; P, progesterone; Ni, nickel.

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
