# Peer review of "The Roles of Sex Hormones in the Course of Atopic Dermatitis"

_ijms, 2019, doi:10.3390/ijms20194660_

Round 1

Reviewer 1 Report

In this manuscript the authors investigate how the expression of sex hormones (i.e.  estrogen, progesterone, androgens, and DHEA) affect immune responses and skin barrier. They relate these observations back to atopic dermatitis (AD), proposing potential hypotheses for why there are slight sex-based differences in the disease pre and post puberty. Overall, this manuscript could be very helpful to the research community working on AD.

The most interesting part of the manuscript is how the authors  summarize the literature on sex hormones and immune responses in the test as well as into comprehensive tables that could be very useful for others in the field. The summary on skin barrier defects and  sex hormone changes was also particularly compelling. However, some of the conclusions in the AD sections were a bit overstated based on the current data.

Comment 1: Please carefully edit the manuscript for grammar errors, also some of the statements are sometime difficult to follow because English construction. There are typos in figures and text and unconventional usage of terms. Few examples below:

-Figure 1 ceramid synthesis -> “ceramide”.

-The authors also state that AD is a chronic inflammatory dermatosis: Dermatosis means a non-inflammatory skin disease i.e. not a dermatitis.

- The authors mention type 2 helper T cell and go on to use this terminology. It is either type 2 immunity or T helper 2 cell, both of which have different meanings in the immunology field.

Comment 2: Figure 2 could be updated /simplify to only contain pertinent information. Specifically, the authors only discuss the different levels of hormones during the menstrual cycles so ovarian histology, body temp, endometrial histology are all irrelevant information.

Comment 3: The tables are the strongest part of this manuscript yet suffer from poor formatting and at times are incomprehensible. Maybe too large and difficult to read. Will recommend reformatting and/or simplify it.

-Undefined terms are used: Table 1 , what does this mean? That there is a range of effects from moderate stimulation to strong inhibition?

-The “Total activity or Total effects” row at the end of each table is unnecessary and confusing.

-The columns and headings need to be justified correctly. There is a lot of confusion with what lines up with what. This especially occurs when the table goes onto another page and the headings are not there anymore. Add them to every page.

                *May be easier to color code things that are increased or decreased

                *Should group similar observations together and put multiple references at the end so there are less rows. Example Table 2 RORγt, vivo, mice Th17 increased – Refs 19 and 24 are showing same thing but in different lines, which is not necessary.

Comment 4: In the introduction authors mention AD skin barrier defect as only limited to ceramide and FLG mutation. It might be important to mention that other barrier defects have been describe, including Tight Junction. This could be particular relevant as the effect of sex hormones have been shown on TJ structure in several epithelial models (over 160 references when searching : Tight junction and sex hormones in pubmed). Maybe just mentioning this could be relevant for other in the field.

Comment 5: Th17 cells have been also shown in AD skin, for completeness this could be added in Fig 1

Comment 6: Lines 88-89: Author stated that in general men are more Th1 and female Th2 skewed; are there any data showing that women have higher Th2 markers (e.g. IgE or TARC) in atopic disorder (e.g. AD, asthma)?

Comment 7: Section 5: Will suggest revise this section,  not sure fully summarize our understanding on intrinsic AD. Not all cases of intrinsic AD are driven by Nickel sensitization. Also previous studies have suggested that Intrinsic AD shows similar Th2 immune response compared to extrinsic ( J Allergy Clin Immunol. 2013 Aug;132(2):361-70.) Will suggest keeping  this section short, more data focus and less “speculative” .

Comment 8: some references are missing throughout the manuscript.  Also, will suggest being a bit more clear in the statement on what is a speculation and what has been shown. For examples:

 Line 137: women are more susceptible to DHEA, is this the authors hypothesis or has been proved? In this case reference should be included. Higher level in the blood might not necessary indicate more activity, as other factors (e.g. receptor and downstream pathways) might play a role as well in the function

Section 9, line 350: Will suggest revise the conclusive statement instead “indicating” author could say that one can speculate that…but confirmatory studies are needed.

Line 186: Ref is missing

Lines 286-288: Ref are missing

Matthew Brewer post-doctor at the University of Rochester Medical Center, Rochester, has worked with me on this manuscript review. Matthew Brewer has given many constructive comments.

Author Response

Responses to Reviewer 1

Comments and Suggestions for Authors

In this manuscript the authors investigate how the expression of sex hormones (i.e.  estrogen, progesterone, androgens, and DHEA) affect immune responses and skin barrier. They relate these observations back to atopic dermatitis (AD), proposing potential hypotheses for why there are slight sex-based differences in the disease pre and post puberty. Overall, this manuscript could be very helpful to the research community working on AD.

The most interesting part of the manuscript is how the authors summarize the literature on sex hormones and immune responses in the test as well as into comprehensive tables that could be very useful for others in the field. The summary on skin barrier defects and sex hormone changes was also particularly compelling. However, some of the conclusions in the AD sections were a bit overstated based on the current data.

 Comment 1: Please carefully edit the manuscript for grammar errors, also some of the statements are sometime difficult to follow because English construction. There are typos in figures and text and unconventional usage of terms. Few examples below:

-Figure 1 ceramid synthesis -> “ceramide”.

Response: The spelling is corrected.

-The authors also state that AD is a chronic inflammatory dermatosis: Dermatosis means a non-inflammatory skin disease i.e. not a dermatitis.

Response: That is corrected to ‘a chronic inflammatory skin disease’.

- The authors mention type 2 helper T cell and go on to use this terminology. It is either type 2 immunity or T helper 2 cell, both of which have different meanings in the immunology field.

Response: T helper 2 cell (Th2) is used in the revised manuscript.

Comment 2: Figure 2 could be updated /simplify to only contain pertinent information. Specifically, the authors only discuss the different levels of hormones during the menstrual cycles so ovarian histology, body temp, endometrial histology are all irrelevant information.

Response: As recommended by the reviewer, ovarian histology, body temperature, and endometrial histology are deleted. In the legend of this figure, we provided the link to the license of this figure.

Comment 3: The tables are the strongest part of this manuscript yet suffer from poor formatting and at times are incomprehensible. Maybe too large and difficult to read. Will recommend reformatting and/or simplify it.

-Undefined terms are used: Table 1, what does this mean? That there is a range of effects from moderate stimulation to strong inhibition?

Response: ↑~↓means the stimulatory or inhibitory effects of estrogen on Th1 and Th17 activities, dependent on the concentration, tissue, or disease context. This is described at the bottom of this table.

-The “Total activity or Total effects” row at the end of each table is unnecessary and confusing.

Response: These are deleted.

-The columns and headings need to be justified correctly. There is a lot of confusion with what lines up with what. This especially occurs when the table goes onto another page and the headings are not there anymore. Add them to every page.

                *May be easier to color code things that are increased or decreased

                *Should group similar observations together and put multiple references at the end so there are less rows. Example Table 2 RORγt, vivo, mice Th17 increased – Refs 19 and 24 are showing same thing but in different lines, which is not necessary.

Response: We have reformatted Tables 2-5, reduced the row number, and put together the similar effects on Th1, Th2, Th17 or Treg activities in each table.

Comment 4: In the introduction authors mention AD skin barrier defect as only limited to ceramide and FLG mutation. It might be important to mention that other barrier defects have been describe, including Tight Junction. This could be particular relevant as the effect of sex hormones have been shown on TJ structure in several epithelial models (over 160 references when searching : Tight junction and sex hormones in pubmed). Maybe just mentioning this could be relevant for other in the field.

Response: Though over 160 references are revealed when searching: Tight junction (TJ) and sex hormones in pubmed, there are only two papers when searching TJ and sex hormones and skin (refs 88 and 89). Ref 89 has already been discussed in the original manuscript. Ref 88 is newly cited and discussed in the 4th paragraph of section 3.

Comment 5: Th17 cells have been also shown in AD skin, for completeness this could be added in Fig 1

Response: We have added the description on Th17 in this figure.

Comment 6: Lines 88-89: Author stated that in general men are more Th1 and female Th2 skewed; are there any data showing that women have higher Th2 markers (e.g. IgE or TARC) in atopic disorder (e.g. AD, asthma)?

Response: This statement in section 2.1 is according to the conclusion in ref 18: Roved, J.; Westerdahl, H.; Hasselquist, D. Sex differences in immune responses: Hormonal effects, antagonistic selection, and evolutionary consequences. Horm Behav 2017, 88, 95-105, doi:10.1016/j.yhbeh.2016.11.017. This article searched plenty of papers regarding the effects of sex hormones on Th2 and Th1 activities. There are no definitive data showing that women have higher Th2 markers (e.g. IgE or TARC) in atopic disorder (e.g. AD, asthma).

Comment 7: Section 5: Will suggest revise this section, not sure fully summarize our understanding on intrinsic AD. Not all cases of intrinsic AD are driven by Nickel sensitization. Also previous studies have suggested that Intrinsic AD shows similar Th2 immune response compared to extrinsic ( J Allergy Clin Immunol. 2013 Aug;132(2):361-70.) Will suggest keeping this section short, more data focus and less “speculative”.

Response: We have newly cited this paper (ref 100), and described that intrinsic AD patients show Th2 activity comparable to that of extrinsic AD patients in section 5. We have also described that metals might act as the main allergens in intrinsic AD though the other agents might also work in its pathogenesis in the last half of this section. Moreover, we also described that not all intrinsic AD patients show Ni allergy in this section.

Comment 8: some references are missing throughout the manuscript. 

Response: We have inserted the references in several portions of the revised manuscript: section 2.1 (refs 17, 18, 19, 20); first paragraph of section 3 (refs 77, 78, 79).

Also, will suggest being a bit more clear in the statement on what is a speculation and what has been shown.

Response: We have revised the statement at the portions of speculation: last sentences of section 8 and 9.

For examples:

 Line 137: women are more susceptible to DHEA, is this the authors hypothesis or has been proved? In this case reference should be included. Higher level in the blood might not necessary indicate more activity, as other factors (e.g. receptor and downstream pathways) might play a role as well in the function

Response: That is the hypothesis of authors according to that in ref 20. This portion in section 2.4 is thus rephrased as recommended by the reviewer above.

Section 9, line 350: Will suggest revise the conclusive statement instead “indicating” author could say that one can speculate that…but confirmatory studies are needed.

Response: This portion in section 9 is rephrased as recommended by the reviewer.

Line 186: Ref is missing

Response: Refs 78 and 79 are inserted in section 3.

Lines 286-288: Ref are missing

Response: Refs 113, 114, and 18 are inserted in the second paragraph of section 6.

Finally, thank you very much for the review and important comments on our manuscript.

Submission Date

16 July 2019

Date of this review

07 Aug 2019 19:23:29

Reviewer 2 Report

In this article authors reviewed effects of sex hormones in the course of atopic dermatitis. They reviewed effect of sex hormones in general immune response and the skin barrier and they proposed hypothesis that hormonal regulation at different age and gender with accent on adolescence/adulthood might be responsible for different prevalence of intrinsic/extrinsic AD in females and males.

This is well written review and from the data presented, hypothesis seems to be plausible.

I have only minor corrections spoted in line 35 (clinocopathological -->> clinicopathological) and 278 (sufatase­ -->> sulfatase).

Author Response

Responses to Reviewer 2

Comments and Suggestions for Authors

In this article authors reviewed effects of sex hormones in the course of atopic dermatitis. They reviewed effect of sex hormones in general immune response and the skin barrier and they proposed hypothesis that hormonal regulation at different age and gender with accent on adolescence/adulthood might be responsible for different prevalence of intrinsic/extrinsic AD in females and males.

This is well written review and from the data presented, hypothesis seems to be plausible.

I have only minor corrections spoted in line 35 (clinocopathological -->> clinicopathological) and 278 (sufatase­ -->> sulfatase).

Response: We have corrected these spelling errors in sections 1 and 6.

Thank you for the review and comments.

Submission Date

16 July 2019

Date of this review

13 Sep 2019 16:18:10

Reviewer 3 Report

The article aims to present the role of sex hormones on atopic dermatitis (AD), by reviewing supporting evidence in medical literature. The authors present the effect of sex hormones on immune response, in an organised manner, then approach the effects of sex hormones on skin barrier function. Moreover the authors launch hypotheses regarding the involvement of sex hormones in the sex differences observed in patients with AD in the clinical setting.

The article brings an updated and refreshed perspective, the information is presented in a structured manner, and the work is thoroughly documented. Molecular interplays are quite well described, in an accesible manner.

Lines 35-37 – please reflect upon the use of the definite article “the” in this phrase;

Line 42- the same as above

Line 50 – please reconsider the use of uderlining of “thymic stromal lymphopoietin” in figure legend

Line 59- “the secretion of sex hormones from the reproductive organs...” – please be more specific regarding the exact source of hormones

Line 2,3, 42, 83 – subtitles need adequate numbering, in order to give an overall better organised impression and to facilitate reading of the article.

Line 90 –“ Androgens, dihydrotestosterone (DHT) and testosterone, are synthesized” please reconsider the use of comma

Line 92- “Testosterone is the most concentrated androgen in adult male serum, while DHT is present at one-tenth the concentration of testosterone though DHT is more potent than testosterone” – please rephrase

Lines 166-169 “In 2,4-dinitrochlorobenzene (DNCB)-induced AD model mice, oral or topical

167 DHEA attenuated eosinophil and mast cell infiltration into ear skin challenged with DNCB, and reduced serum IL-4 and IgE levels, and reduced IL-4 and IL-5 production while increased IFN- production in splenocytes [67].” –please rephrase as to be clearer.

Line 276 “Before puberty, children are mostly devoid of the influence by sex hormones, and the effects” –please rephrase, considering that sexual hormones, even in low concentration are also present in children.

Please also take into consideration the following article:

J Dermatol Sci. 2015 Nov;80(2):116-23. doi: 10.1016/j.jdermsci.2015.09.005.

Atopic dermatitis-like skin lesions with IgE hyperproduction and pruritus in KFRS4/Kyo rats.

by Kuramoto T, Yokoe M, Tanaka D, Yuri A, Nishitani A, Higuchi Y, Yoshimi K, Tanaka M, Kuwamura M, Hiai H, Kabashima K, Serikawa T.

The article is somewhat dense; it would benefit from proper title/subtitle numbering, leading to a more structured appearance that would enhance the reader’s experience.

The authors are advised to address this aspect, to improve reader’s orientation in the article.

Also, please check grammar and spelling throughout the article.

Overall, by summing up a large quantity of evidence-based studies, the article might be a useful contribution to the journal. I recommend the article to be published, after all suggested changes are taken into consideration by authors.

Author Response

Response to Reviewer 3

Comments and Suggestions for Authors

The article aims to present the role of sex hormones on atopic dermatitis (AD), by reviewing supporting evidence in medical literature. The authors present the effect of sex hormones on immune response, in an organised manner, then approach the effects of sex hormones on skin barrier function. Moreover the authors launch hypotheses regarding the involvement of sex hormones in the sex differences observed in patients with AD in the clinical setting.

The article brings an updated and refreshed perspective, the information is presented in a structured manner, and the work is thoroughly documented. Molecular interplays are quite well described, in an accesible manner.

Lines 35-37 – please reflect upon the use of the definite article “the” in this phrase;

Line 42- the same as above

Response: These portions in section 1 are rephrased as recommended by the reviewer.

Line 50 – please reconsider the use of uderlining of “thymic stromal lymphopoietin” in figure legend

Response: I do not know the reason why this term is underlined in the pdf automatically constructed independent on my will. The term in figure 1 legend in my original word document is not underlined.

Line 59- “the secretion of sex hormones from the reproductive organs...” – please be more specific regarding the exact source of hormones

Response: We have specified the organs in the 3rd paragraph of section 1.

Line 2,3, 42, 83 – subtitles need adequate numbering, in order to give an overall better organised impression and to facilitate reading of the article.

Response: We have renumbered the subtitles as follows: 2.1 General tendency; 2.2 Female hormones; 2.2.1 Estrogens; 2.2.2 Progesterone; 2.3 Androgens; 2.4 DHEA.

Line 90 –“ Androgens, dihydrotestosterone (DHT) and testosterone, are synthesized” please reconsider the use of comma

Line 92- “Testosterone is the most concentrated androgen in adult male serum, while DHT is present at one-tenth the concentration of testosterone though DHT is more potent than testosterone” – please rephrase

Response: We have rephrased these portions in section 2.3.

Lines 166-169 “In 2,4-dinitrochlorobenzene (DNCB)-induced AD model mice, oral or topical

167 DHEA attenuated eosinophil and mast cell infiltration into ear skin challenged with DNCB, and reduced serum IL-4 and IgE levels, and reduced IL-4 and IL-5 production while increased IFN- production in splenocytes [67].” –please rephrase as to be clearer.

 Response: It is topical DNCB. This portion is rephrased in the 2nd paragraph of section 2.4.

Line 276 “Before puberty, children are mostly devoid of the influence by sex hormones, and the effects” –please rephrase, considering that sexual hormones, even in low concentration are also present in children.

Response: This portion in the first paragraph of section 6 is rephrased as recommended by the reviewer.

Please also take into consideration the following article:

J Dermatol Sci. 2015 Nov;80(2):116-23. doi: 10.1016/j.jdermsci.2015.09.005.

Atopic dermatitis-like skin lesions with IgE hyperproduction and pruritus in KFRS4/Kyo rats.

by Kuramoto T, Yokoe M, Tanaka D, Yuri A, Nishitani A, Higuchi Y, Yoshimi K, Tanaka M, Kuwamura M, Hiai H, Kabashima K, Serikawa T.

The article is somewhat dense; it would benefit from proper title/subtitle numbering, leading to a more structured appearance that would enhance the reader’s experience.

The authors are advised to address this aspect, to improve reader’s orientation in the article.

Response: Thank you very much for notifying us of this important paper. We have newly cited this paper (ref 13) and discussed in the 3rd paragraph of section 1. This paper shows female preponderance of AD-like dermatitis in KFRS4/Kyo rats possibly after puberty, and indicates the effects of sex hormones in the female preponderance.  

Also, please check grammar and spelling throughout the article.

Response: We have carefully checked grammatical and spelling errors in the manuscript, and have corrected these errors. 

Overall, by summing up a large quantity of evidence-based studies, the article might be a useful contribution to the journal. I recommend the article to be published, after all suggested changes are taken into consideration by authors.

Response: The portions suggested by the reviewer are revised.

Thank you very much for the review and important comments.

Submission Date

16 July 2019

Date of this review

13 Sep 2019 17:41:08